REGISTERED REPORT PROTOCOL

# Is reporting quality in medical publications associated with biostatisticians as co-authors? A registered report protocol

Ulrike Held[1]*, Klaus Steigmiller[1], Michael Hediger[2], Martina Gosteli[3], Kelly A. Reeve[1], Stefanie von Felten[1], Eva Furrer[1]

1 Department of Biostatistics at Epidemiology, Biostatistics, and Prevention Institute, University of Zurich, Zurich, Switzerland, 2 Institute of Mathematics, University of Zurich, Zurich, Switzerland, 3 Main Library, University of Zurich, Zurich, Switzerland

* ulrike.held@uzh.ch

## Abstract

### Background

Quality in medical research has recently been criticized for being low, especially in observational research. Methodology is increasingly difficult, but collaboration between clinical researchers and biostatisticians may improve research and reporting quality. The aim of this study is to quantify the value of a biostatistician in the team of authors.

### Methods

Single-center, retrospective observational study following the STROBE reporting guidelines. We will systematically review all medical publications with biostatisticians from our center as co-authors or authors and review corresponding papers without biostatisticians from our center during the same time range. We will compare aspects of reporting quality, overall and for the three study types observational, randomized trial, and prognostic separately.

### Discussion

We anticipate that the results of the study will raise awareness of the importance of high methodological quality, as well as appropriate reporting quality in clinical research.

### Conclusion

Our study will have a direct impact on our center by making each of us more aware of the reporting guidelines for various research designs. This in turn will enhance reporting quality in future research with our involvement. Our study will also raise awareness of the important role that biostatisticians play in the design and analysis of health research projects.

**Data Availability Statement:** This is a registered report protocol. The data collected in this research project will be made available upon finalization of the study together with corresponding statistical programming code and Shiny Apps.

**Funding:** EF and MH received funding from the Center for Reproducible Science (CRS) at the University of Zurich. The funder had and will not have a role in study design, data collection and analysis, decision to publish, or preparation of the manuscript.

**Competing interests:** The authors have declared that no competing interests exist.

## Introduction

In recent years, the methodological and reporting quality of biomedical research has been extensively discussed. In times when the mere number of published papers seems to pave the academic career path for health research scientists, there is increasing motivation to publish for the sake of another paper rather than for the sake of scientific advancement [1]. In Switzerland, the government paid 0.9% of the gross domestic product for research and development in 2016, and more than 20% of all publications in Switzerland between 2011 and 2015 were in the field of "clinical medicine" [2]. The government is paying large amounts of money for medical research, but as Doug Altman said, "To maximise the benefit to society, you need to not just do research, but do it well." [3]. Some medical journals like JAMA, BMJ, PLOS Medicine, The Lancet, and BMC Medicine, among others, identified this problem many years ago, sparking groups of methodological experts to discuss reliability, perceived value, and reporting quality of research. Simera and co-authors discussed the deficiencies in reporting medical research, making it difficult to assess "how the research was conducted" and to "evaluate the reliability of presented findings" [4], thus hampering its utilization in clinical practice. To maximize the value from funded research, so-called reporting guidelines were developed to foster good research practices among clinical scientists, which are now organized within the EQUATOR network (www.equator-network.org). The first evidence-based recommendation on how to report on results of randomized controlled trials (RCTs) was the CONSORT guideline, published in its first version in 1996. Since then, the CONSORT guideline has been revised [5, 6] and further reporting guidelines have been developed. These address common study types in medical research, with STROBE guidelines for observational studies [7], and TRIPOD guidelines for reporting of prediction or prognostic models [8] among the most frequently used.

Despite a large effort put into the development and publication of these reporting guidelines by medical journals, the quality standards in medical publications are often not met. An indication for this is e.g. increasing retraction rates observed in high rank medical journals [9]. In recent studies by Dechartres et al. [10], Zuniga-Hernandez et al. [11] and Sharp et al. [7] the authors found that medical journals should enforce the use of the reporting guidelines more strictly. Among other reasons, limited understanding of research methodology and biostatistics among clinicians might be a reason for poor conduct and reporting quality [12]. It is common practice in clinical research for clinical authors to use unsuitable methods on complex data sets rather than seek advice from methodological experts. Interdisciplinary teams of medical researchers and biostatisticians, however, would be expected to produce higher quality publications [13] and could thus increase public benefit from medical research.

Authorship guidelines at our center (University of Zurich, Switzerland) are based on fulfilling the criteria of Swiss Academies of Arts and Sciences, by (1) making a substantial contribution to the planning, execution, evaluation and supervision of research; (2) involvement in writing the manuscript; and (3) approving the final version of the manuscript. Sometimes, a collaborating biostatistician is mentioned in the acknowledgements section. However, only by being a co-author the biostatistician can take the responsibility to write parts of the manuscript (e.g. statistical methods, results), critically revise the manuscript, and approve the final version.

The objective of our study is to quantify the association between biostatistical co-authorship and the quality of reporting in medical publications, based on a set of pre-defined quality criteria. A secondary objective is to identify study types with methodological knowledge gaps and to promote awareness of the complexity in these areas among clinical researchers as well as among methodologists.

## Methods

The study is a retrospective, single-center observational cohort study, conducted at the University of Zurich (UZH) and its University Hospital (USZ). The group of 13 consulting academic biostatisticians and statisticians from the Epidemiology, Biostatistics and Prevention Institute, and the Institute of Mathematics, both of University of Zurich will be referred to as "biostatisticians" in the following.

### Selection of exposed and non-exposed publications cohorts

Two groups of publications will be compared, and these will be referred to as "exposed" and "non-exposed", corresponding to their "exposure" to a biostatistician of our group in their team. The two cohorts are matched for location, publication year, and study type, with a frequency matching approach [14]. To define the group of exposed publications, all medical research publications from PubMed between 2017 and 2018 with at least one of the biostatisticians as main or co-author was retrieved on Dec 9, 2019, with a search term as specified in S1 Appendix. Methodological publications of these biostatisticians were not included. Publications had to be in English language.

To define the group of "non-exposed" publications for comparison, all medical research publications found in PubMed between 2017 and 2018, with affiliation University of Zurich (UZH) or University Hospital Zurich (USZ) or any of the affiliated university hospitals for first and / or second author were extracted on Dec 16, 2019 (S1 Appendix). The non-exposed publications do not have any of the biostatisticians of our group on the author list. The full list of affiliations considered can be found in S1 File. This large number of publications will be used in a random but replicable order—aiming to remove potential chronological ordering or any other systematic ordering while adhering to high standards of reproducibility.

### Categorization into study types

For each of the exposed publications, the study type was determined, and the subset of all observational studies, RCTs and prognostic studies was evaluated further regarding reporting quality. Categorization into study type was performed by the group of biostatisticians. For most publications, the authors themselves determined the study type. For some publications, the authoring biostatistician had left our group, and thus the study type was categorized independently and in duplicate by two authors (UH, EF). After consensus on study type was reached, record count for each study type for each publication year was obtained.

As the number of potential non-exposed publications was much larger, the categorization of these publications into observational studies, RCTs and prognostic studies was performed in the above described random but replicable order until the numbers of non-exposed publications of these study types matched the corresponding number of exposed publications per year. Categorization was performed independently and in duplicate by the authors UH and MH (for papers published in 2017), and by EF and MH (for 2018). Any discrepancies were resolved by discussion and third-party arbitration (KS). This set of publications was considered the non-exposed group.

### Selection of items from reporting guideline to measure reporting quality

For each of these three study types, a set of six items measuring reporting quality have been identified by reaching group consensus in the group of biostatisticians at our university. The quality criteria are based on the reporting guidelines STROBE, CONSORT and TRIPOD and

reflect characteristics of a publication, that are especially important for judging the validity of the results.

## Specification of the reporting quality items

The reporting guideline items chosen for the ratings represent the following general quality dimensions for all three study types

1. variable specification,

2. how study size was arrived at,

3. missing data,

4. statistical methods,

5. precision of results,

6. whether the corresponding reporting guideline (i.e. STROBE, CONSORT, TRIPOD) was mentioned.

The rating of publications regarding the six items for each of the study types was operationalized and piloted, such that they could be used efficiently and robustly to rate each publication's reporting quality. Each dimension has different possible answer categories, resulting in the overall rating. Details of the operationalization can be found in S2 File.

## Outcomes and quantification of reporting quality

The primary outcome of this study is the reporting quality of exposed and non-exposed publications with respect to the six quality dimensions. The primary outcome will be assessed in blinded fashion and in duplicate by two raters. Blinding will be guaranteed by removing author names, affiliation lists, journal name, corresponding author name, author contributions, date, acknowledgements, references and DOI from every PDF. Discrepancies in the rating will be resolved by a third rater's judgement and discussion until consensus is reached.

To quantify reporting quality, we assign equal weights to each answer category. In that case, the range of the quality dimension score per publication would be between 0 (representing lowest possible reporting quality), and 11 (indicating highest possible reporting quality).

The secondary outcome of this study is the number of citations of both the exposed and the non-exposed publications at a fixed date.

## Outcome rating and rater training

The outcome rating and its operationalization was developed and discussed among four authors (UH, KS, MH, EF), until a consensus was reached. After operationalization was finalized, the resulting questions for each study type were programmed to be shown in an R shiny app, which underwent a quality review and testing period. The raters of exposed and non-exposed publications will be instructed and trained by using vignette publications for calibration. These vignette publications will be related to the study, but published in 2019, and will be rated with scrutiny by the authors of this study. These vignettes contain examples of poor and good reporting quality. The raters will be trained by authors of this study, but the authors of this study will not be involved in the reporting quality rating themselves. The raters will be obliged to rate the reporting quality based on the blinded PDFs alone, and not to use additional information from the internet while doing so. Ratings will be performed in blinded fashion, meaning that the raters will be unaware of the classification of publications as exposed or

non-exposed and of authors on the publications. Ratings will be performed in duplicate, and any discrepancies will be resolved by discussion until consensus is reached.

## Sample size considerations

With the given sample size of 95 exposed and 95 non-exposed publications in 2017 and 2018, at a significance level of 5% and with a power of 80%, an effect size of 0.41 (Cohen's *d*) could be detected using a two-sample 2-sided t-test with equal variances. The effect size would be considered a medium effect size.

## Data management

Categorization into one of the three study types was performed with the help of a specifically programmed R shiny app, in which title and abstract, as well as link to the full text was provided.

Reporting quality rating will be performed again using a shiny app. Electronic records of the reporting quality ratings before and after consensus will be made available. The use of shiny apps in this research guarantees highly reliable data entry.

## Risk of bias

Risk of detection bias will be addressed with blinded outcome ratings. Risk of selection bias will be addressed with reproducible, random subsampling of PubMed publications from medical publications with UZH/USZ affiliation for the group of non-exposed publications. The results of this study could be confounded by indication, if more complex projects were brought to our attention whereas less complex projects were addressed by the clinicians without asking for help from an academic biostatistician.

## Statistical methods and programming

Statistical methods for the primary outcome will include visualization of results with spider plots for reporting quality dimension according to study type, and in total across study types. For both analyses, the raw primary outcome, and the mean difference in reporting quality rating between exposed and non-exposed publications will be reported with a 95% confidence interval. The two-sample t-test or the two-sample Wilcoxon test will be used for group comparisons, the decision to use a parametric or a non-parametric test will be based on visual inspection of histograms and corresponding QQ-plots. Descriptive statistics of the primary outcome for all exposed and non-exposed publications in subgroups of study types will be shown in a table.

Number of citations on a specified date will be collected and compared between the groups of exposed and non-exposed publications, again with either t-tests or Wilcoxon tests. The decision will be based upon visual inspection of histograms and QQ-plots. Between-group differences will be reported with 95% confidence intervals. Confidence intervals will be calculated based on the t-distribution assuming equal variances or with Hodges-Lehman estimate based on the Wilcoxon test, corresponding to the statistical test chosen for each outcome.

For assessing the level of agreement of reporting quality between the two raters, Cohen's kappa (κ) values will be estimated, again with 95% confidence intervals. κ values will be weighted with squared weights. Low levels of agreement would indicate that the ratings are complex, and that third-party arbitration was required in many publications. Again, these κ values will be reported overall and in subgroups of study types. Dependence of κ values on marginals will be taken into consideration during interpretation of results.

All statistical programming will be performed with R, in combination with dynamic reporting. Statistical programming included downloading all potential non-exposed publications, random reordering, development of a shiny app for categorization of the publications, development of a shiny app for reporting quality rating of the publications, as well as statistical methods for comparison of the exposed and non-exposed groups, and its graphical display.

The results of the study will be reported according to the STROBE guidelines [15]. All data (anonymized) and code, together with the shiny apps will be made available upon publication.

## Current status

The electronic search was performed in December 2019, and it resulted in 132 publications with biostatisticians from our group as authors or co-authors, in 2017 and 2018. Of these studies, 77 were observational studies, six were RCTs, and 12 were prognostic studies, 95 in total. The remaining papers were of other study types as the ones considered in this study. In the group of potential non-exposed publications, there were 3559 papers with suspected affiliation University of Zurich / University Hospital Zurich. After removal of unsuitable affiliations, there were 3420 papers with a first or second author having an affiliation at UZH/USZ in 2017 and 2018. Details can be found in Fig 1.

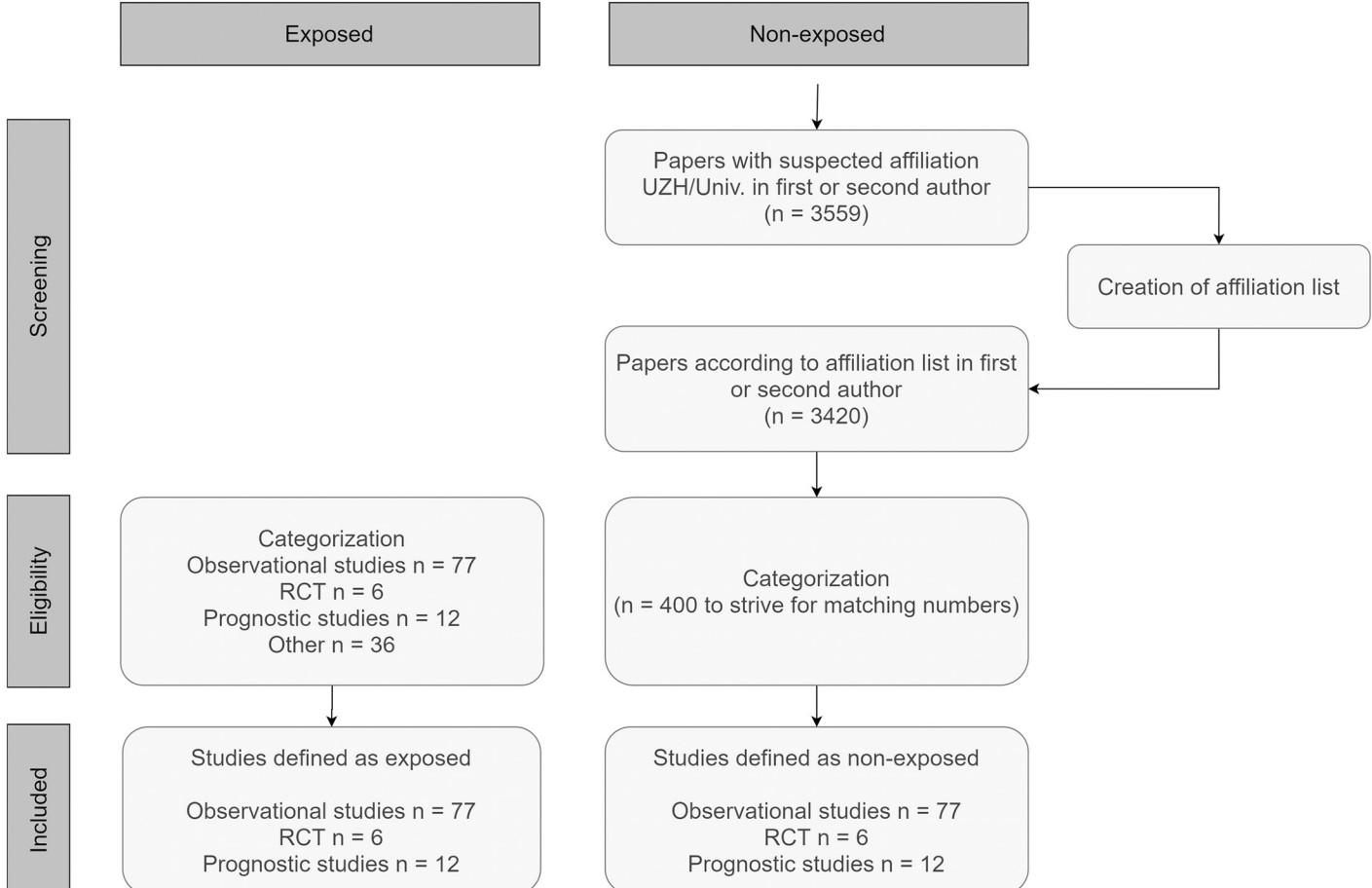

**Fig 1. Flowchart of study selection.** The flowchart shows selection of exposed publications and non-exposed publications, after creation of an affiliation list, and categorization into one of the three study types observational study, RCT, and prognostic study.

## Timeline

While the registered report protocol is under review, we will simultaneously upload the submitted version on the Open Science Framework (https://osf.io/), to raise awareness, receive additional comments, and inform potential raters about the planned research. Upon successful registration of the protocol, requested changes to study design will be made, the raters will be appointed, and rater training will be initiated. Ratings of reporting quality are planned to start in fall 2020. Data will be available in December 2020, and data analysis as well as writing of the manuscript will take place in early 2021.

## Discussion

The study was designed to evaluate and quantify the impact of a biostatistician in an academic setting on the reporting quality of medical publications. From our perspective, the topic is relevant and raises the awareness at our center as well as in a broader audience. Related publications have also addressed this topic recently [16].

This study has several strengths. First, it takes a very systematic and reproducible approach. The approach could easily be adapted to other publication years and universities. Rating of reporting quality is done in a blinded fashion and in duplicate. However, potential sources of bias, such as confounding by indication, may affect our results. This might be the case if biostatisticians were asked for advice only in the more complex research projects, whereas less complex projects were completed without a biostatistician. In that case, it would be more difficult to obtain high scores for reporting quality in exposed publications as compared to in non-exposed publications, resulting in an underestimated between-group difference. Another limitation of this study might be our consensus decision for the reporting items addressed, as well as the weight they were given in the ratings. These might represent our personal views and will be opened for discussion. Results will depend on these choices and need discussion. Another limitation of our study is that we will not account for other biostatisticians (outside of our group) potentially involved in the research in the non-exposed publications. This limitation will lead to a conservative estimate of the association of a biostatistician on reporting quality. Finally, there is potential for misclassification of publications with a biostatistician responsible for the analysis, but with a mention only in the acknowledegements. However, in such situations, the biostatistician would not get the chance to critically revise the manuscript and to approve the final version—which could nevertheless lead to low reporting quality or even false interpretations of results.

As a consequence of this research, awareness of low reporting quality and guidelines to enable improved reporting will be raised in our centre. It would provide quantified evidence for the need of implementing higher quality reviews by journals and to increase biostatistics knowledge. It could also reveal that if biostatisticians are members of medical research groups, involved at early stages of research projects, rather than just consulting biostatisticians or data analysts, this would generally increase the quality, conduct, and reporting of medical research. Study design is intended to facilitate the use of our approach in other academic centres with biostatistics units involved in medical research. A certain number of publications were not addressed in this manuscript. These were studies with different study types, such as systematic reviews and meta-analyses, study protocols and pre-clinical studies. We aim to address these study types in future research.

Our research has implications for practice in the sense that frequent discussions about the necessity and importance of the reporting items fostered awareness in our group of biostatisticians. Benchmarking in future years against the findings of this study will take place and

already now made the biostatisticians at our centre insist on using the guidelines in their most recent health research publications.

## Conclusion

Our study will have a direct impact on our center by making each of us more aware of the reporting guidelines for various research designs. This in turn will enhance reporting quality in future research with our involvement. Our study will also raise awareness of the important role that biostatisticians play in the design and analysis of health research projects.

## Supporting information

**S1 Appendix. Search string and date of search.** The file contains the search string for exposed and non-exposed publications for use in PubMed.
(PDF)

**S1 File. Affiliation list.** The affiliation list was generated from the electronic search, with indication of included and excluded affiliations.
(PDF)

**S2 File. Questionnaire for assessment of reporting quality.** The questionnaire is intended to be used for rating the reporting quality in each study type.
(PDF)

## Acknowledgments

We thank Steffi Muff, Leonhard Held, Sarah Haile, Julia Braun, Gilles Kratzer, Andrea Götschi, and Charlotte Micheloud for contributing to the selection of reporting items. We also express our gratitude to Tina Wünn who was involved in the operationalization.

## Author Contributions

**Conceptualization:** Ulrike Held, Klaus Steigmiller, Kelly A. Reeve, Stefanie von Felten, Eva Furrer.

**Data curation:** Michael Hediger.

**Investigation:** Martina Gosteli, Kelly A. Reeve.

**Methodology:** Ulrike Held, Klaus Steigmiller, Michael Hediger, Martina Gosteli, Kelly A. Reeve, Stefanie von Felten, Eva Furrer.

**Project administration:** Ulrike Held, Klaus Steigmiller, Eva Furrer.

**Software:** Klaus Steigmiller.

**Supervision:** Eva Furrer.

**Writing – original draft:** Ulrike Held.

**Writing – review & editing:** Ulrike Held, Klaus Steigmiller, Michael Hediger, Martina Gosteli, Kelly A. Reeve, Stefanie von Felten, Eva Furrer.

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
