## [Decision Letter · Decision Letter 0]

14 Sep 2020

PONE-D-20-21077

Is Reporting Quality in Medical Publications Associated with Biostatisticians as Co-Authors? A Registered Report Protocol.

PLOS ONE

Dear Dr. Held,

Thank you for submitting your manuscript to PLOS ONE. After careful consideration, we feel that it has merit but does not fully meet PLOS ONE’s publication criteria as it currently stands. Therefore, we invite you to submit a revised version of the manuscript that addresses the points raised during the review process.

Please attend to the minor comments.  All of the reviewers thoroughly found this work important.

We look forward to receiving your revised manuscript.

Kind regards,

Alan D Hutson

Academic Editor

PLOS ONE

Additional Editor Comments:

Please attend to the minor comments. All of the reviewers thoroughly found this work important.

Journal Requirements:

Reviewers' comments:

Reviewer's Responses to Questions

**Comments to the Author**

1. Does the manuscript provide a valid rationale for the proposed study, with clearly identified and justified research questions?

Reviewer #1: Yes

Reviewer #3: Yes

Reviewer #4: Yes

2. Is the protocol technically sound and planned in a manner that will lead to a meaningful outcome and allow testing the stated hypotheses?

Reviewer #1: Yes

Reviewer #3: Yes

Reviewer #4: Yes

3. Is the methodology feasible and described in sufficient detail to allow the work to be replicable?

Reviewer #1: Yes

Reviewer #3: Yes

Reviewer #4: Yes

4. Have the authors described where all data underlying the findings will be made available when the study is complete?

Reviewer #1: Yes

Reviewer #3: Yes

Reviewer #4: Yes

5. Is the manuscript presented in an intelligible fashion and written in standard English?

Reviewer #1: Yes

Reviewer #3: Yes

Reviewer #4: Yes

6. Review Comments to the Author

You may also provide optional suggestions and comments to authors that they might find helpful in planning their study.

Reviewer #1: I have only minor comments.

1) Your use of 'case' and 'control' is non-standard and may confuse some readers. This is the distinction that is used in observational studies where sampling is by outcome. Sometimes these are called retrospective studies but more commonly the preferable term 'case-control' study is used. For example, you might sample cases of lung-cancer and controls who do not have lung cancer. Other types of study classify groups by exposure, not by outcome. For example, you might classify subjects as exposed 'smoker' or not-exposed 'non-smoker'. Such studies are referred to as cohort studies. It is a standard epidemiological teaching that case control studies are good at studying multiple putative causal factors and cohort studies are good at studying multiple possible effects. Here you have the analogue of a cohort study, albeit with papers rather than subjects as the unit of study. I think it would be more usual to refer to the two groups as either exposed or non-exposed (as in an observational study) or 'treated' or possibly 'experimental' versus 'control' as for an interventional study. I agree that this terminology is not ideal here but I worry about what you have chosen. Perhaps all you need to note at the beginning is that 'case' and 'control' are not being used as in many observational studies.

2) I have checked your sample size calculation and find it to be correct provided that you intend to use a two-sample t-test with equal variances to perform a conventional significance test. Is this what you intend? You don't actually say what the intended analysis corresponding to your sample size calculation is.

3) Later you refer to t-tests and Wilcoxon test and say they will be used 'as appropriate' but you don't actually say how you will decide what is appropriate. (Note I am definitely not encouraging you to use a test of Normality to decide between them. Nevertheless this sort of 'as appropriate' statement is something that would not generally be accepted as adequate for a statistical analysis plan in drug development.)

4) Various typos and minor issues

a) L40 'Doug' not 'Dough'

b)L41 This quotation does not sound quite right. Please check it and provide a reference.

c)L50. Suggest 'most frequently used' rather than 'most frequent'

d) L120. 'TRIPOD' not 'TROPID'

e) L158 0.41 is a rather unusual choice of effect size. Is it the case that this is the value that guarantees 80% power for the sample size you intend to use?

f) L179. How will the confidence intervals be calculated? T-test. Hodges-Lehmann Wilcoxon

Reviewer #3: This is a novel and important research idea that is much needed. As statistical techniques increase in sophistication, somewhat due to increased computing power, we can’t expect physicians and practitioners to always know every statistical nuance that could be applied to their research. Reliance on statisticians to guide methodological decision-making should be normalized to advance science in the best, most sophisticated way.

The authors nicely lay out the purpose of this work, especially in the introductory discussion to include biostatisticians on interdisciplinary research teams despite following STROBE check-lists.

My suggestions to improve this report protocol are as follows:

1. Consider re-labeling the “control” publications group as a “comparison” publications group. Since biostatisticians were not randomly assigned to author teams / projects, it is not a true control, but rather a comparison.

2. In the introductory paragraph that describes the comparison / control group, please be explicitly clear that this group is of papers who do not have a statistician listed as an author. This is implied in the introduction, but it could be clearer in this paragraph. (Lines 84-90, page 5).

3. One item to discuss or acknowledge is whether the research center from which you pull these papers has an authorship policy of some sort that would impact whether a biostatistician is added to the authorship list. For example, is there a standard policy of deciding authorship on papers? If yes or no, please describe. Also, is there an informal or collegial policy that this research center follows with reagrds to authorship? For example, if you know that statisticians are always included as authors when they help with methodologies in this research center, that would be good to state in this report protocol as a certainty that the study cases will be correctly classified.

4. Another thought, that may or may not be relevant, but you may wish to address – are statisticians sometimes acknowledged or thanked in papers but not added as authors? If they are thanked and were a part of the research, but not an author on the manuscript which is then classified in the control group, this may represent a mis-classification of that paper.

Otherwise, the proposed methodology that is laid out in this submission is very well-thought and rigorous. I look forward to reading about the results of this important and interesting work.

Reviewer #4: This is a very interesting study. I especially like the use of R and Shiny. Here are some suggested edits.

Intro section, first sentence - if possible, please say a few words on what you mean by quality.

Second sentence - should be "In times when the mere..." Add the word 'the'?

You quote Doug Altman - but it is misspelled. Please add reference if there is one.

'Some medical journals' are mentioned. can you give examples: (e.g., Lancet, ...)

You mention reliability, perceived value, and reporting quality. Please define these briefly here.

The EQUATOR network is mentioned. Please give a reference or website link.

Methods section:

The frequency matching approach is mentioned. Please give a definition and/or reference.

In the fifth paragraph, initials are used. I realized later these are for the authors. Please mention that here.

I liked how you identified the quality measures.

7. PLOS authors have the option to publish the peer review history of their article (what does this mean?). If published, this will include your full peer review and any attached files.

Reviewer #1: **Yes: **Stephen Senn

Reviewer #3: No

Reviewer #4: No

---

## [Author Response · Author response to Decision Letter 0]

5 Oct 2020

Dear Dr. Hutson

Thank you for informing me and my co-authors about the minor revision of the registered report protocol.

We are resubmitting a revised version of the protocol, that was improved by the reviewer suggestions.

All reviewer comments were addressed in our response letter.

Kind regards

Ulrike Held

---

## [Decision Letter · Decision Letter 1]

23 Oct 2020

Is Reporting Quality in Medical Publications Associated with Biostatisticians as Co-Authors? A Registered Report Protocol.

PONE-D-20-21077R1

Dear Dr. Held,

We’re pleased to inform you that your manuscript has been judged scientifically suitable for publication and will be formally accepted for publication once it meets all outstanding technical requirements.

Kind regards,

Alan D Hutson

Academic Editor

PLOS ONE

Additional Editor Comments (optional):

Reviewers' comments:

Reviewer's Responses to Questions

**Comments to the Author**

1. Does the manuscript provide a valid rationale for the proposed study, with clearly identified and justified research questions?

Reviewer #3: Yes

Reviewer #4: Yes

2. Is the protocol technically sound and planned in a manner that will lead to a meaningful outcome and allow testing the stated hypotheses?

Reviewer #3: Yes

Reviewer #4: Yes

3. Is the methodology feasible and described in sufficient detail to allow the work to be replicable?

Reviewer #3: Yes

Reviewer #4: Yes

4. Have the authors described where all data underlying the findings will be made available when the study is complete?

Reviewer #3: Yes

Reviewer #4: Yes

5. Is the manuscript presented in an intelligible fashion and written in standard English?

Reviewer #3: Yes

Reviewer #4: Yes

6. Review Comments to the Author

You may also provide optional suggestions and comments to authors that they might find helpful in planning their study.

Reviewer #3: The authors have nicely addressed all of the reviewers' comments. I recommend the article for publication.

Reviewer #4: The paper is much improved. All of my comments were addressed adequately, and I have no further edits or issues to address. Thank you.

7. PLOS authors have the option to publish the peer review history of their article (what does this mean?). If published, this will include your full peer review and any attached files.

Reviewer #3: **Yes: **Melissa Kovacs

Reviewer #4: No

---

## [Editor Report · Acceptance letter]

29 Oct 2020

PONE-D-20-21077R1 

Is reporting quality in medical publications associated with biostatisticians as co-authors? A registered report protocol 

Dear Dr. Held:

I'm pleased to inform you that your manuscript has been deemed suitable for publication in PLOS ONE. Congratulations! Your manuscript is now with our production department. 

Kind regards, 

on behalf of

Dr. Alan D Hutson 

Academic Editor

PLOS ONE